# Impacts of Climate Change Interacting Abiotic Factors on Growth, *aflD* and *aflR* Gene Expression and Aflatoxin B_1_ Production by *Aspergillus flavus* Strains In Vitro and on Pistachio Nuts

**DOI:** 10.3390/toxins13060385

**Published:** 2021-05-28

**Authors:** Alaa Baazeem, Alicia Rodriguez, Angel Medina, Naresh Magan

**Affiliations:** 1Department of Biology, College of Science, Taif University, P.O. Box 11099, Taif 21944, Saudi Arabia; aabaazeem@tu.edu.sa; 2Department of Animal Science and Food Production, University of Extramadura, Av. de Elvas, s/n, 06006 Badajoz, Spain; aliciarj@unex.es; 3Applied Mycology Group, Environment and AgriFood Theme, Cranfield University, Cranfield MK43 0AL, UK; a.medinavaya@cranfield.ac.uk

**Keywords:** climate change, interacting abiotic factors, *aflD*, *aflR*, aflatoxins, *Aspergillus flavus*, pistachios

## Abstract

Pistachio nuts are an important economic tree nut crop which is used directly or processed for many food-related activities. They can become colonized by mycotoxigenic spoilage fungi, especially *Aspergillus flavus*, mainly resulting in contamination with aflatoxins (AFs), especially aflatoxin B_1_ (AFB_1_). The prevailing climate in which these crops are grown changes as temperature and atmospheric CO_2_ levels increase, and episodes of extreme wet/dry cycles occur due to human industrial activity. The objectives of this study were to evaluate the effect of interacting Climate Change (CC)-related abiotic factors of temperature (35 vs. 37 °C), CO_2_ (400 vs. 1000 ppm), and water stress (0.98–0.93 water activity, a_w_) on (a) growth (b) *aflD* and *aflR* biosynthetic gene expression and (c) AFB_1_ production by two strains *A. flavus* (AB3, AB10) in vitro on milled pistachio-based media and when colonizing layers of shelled raw pistachio nuts. The *A. flavus* strains were resilient in terms of growth on pistachio-based media and the colonisation of pistachio nuts with no significant difference when exposed to the interacting three-way climate-related abiotic factors. However, in vitro studies showed that AFB_1_ production was significantly stimulated (*p* < 0.05), especially when exposed to 1000 ppm CO_2_ at 0.98–0.95 a_w_ and 35 °C, and sometimes in the 37 °C treatment group at 0.98 a_w_. The relative expression of the structural *aflD* gene involved in AFB_1_ biosynthesis was decreased or only slightly increased, relative to the control conditions at elevated CO, regardless of the a_w_ level examined. For the regulatory *aflR* gene expression, there was a significant (*p* < 0.05) increase in 1000 ppm CO_2_ and 37 °C for both strains, especially at 0.95 a_w_. The in situ colonization of pistachio nuts resulted in a significant (*p* < 0.05) stimulation of AFB_1_ production at 35 °C and 1000 ppm CO_2_ for both strains, especially at 0.98 a_w_. At 37 °C, AFB_1_ production was either decreased, in strain AB3, or remained similar, as in strain AB10, when exposed to 1000 ppm CO_2_. This suggests that CC factors may have a differential effect, depending on the interacting conditions of temperature, exposure to CO_2_ and the level of water stress on AFB_1_ production.

## 1. Introduction 

Pistachio nuts (*Pistacia vera* L.) are one of the most economically important tree nut crops in countries such as the USA, Iran, Southern Europe and China. Of these, Iran and the USA are probably the major producers and exporters of pistachio nuts for direct consumption and for processing [1]. The quality and safety of pistachio nuts for human consumption have been compromised by the infection of the ripening nuts, especially pre-harvest, when early splitting of the shells can allow colonization by members of the *Aspergillus* section *Flavi* group, predominantly *A. flavus.* In addition, poor drying and storage practices can also exacerbate infection by *A. flavus* post-harvest [2,3]. Such infection results in significant contamination with aflatoxin B_1_ (AFB_1_), a class 1a carcinogen [4]. This has resulted in many countries importing pistachio nuts, including the EU, to have strict legislative maximum allowable contamination levels with AFB_1_ or total aflatoxins (AFs). Thus, Iranian pistachio nuts were excluded from importation into the EU, because batches consistently exceeded the prevailing legislative limits for AFs [5]. This required the development of an effective Hazard Analysis Critical Control Plan (HACCP) with appropriate Critical, Control Points in this chain focused on more efficient harvesting and post-harvest management of pistachios from Iran to meet the necessary legislative limits. However, because they are rich in lipids, they can reabsorb moisture from the atmosphere during the post-harvest phase, allowing for further contamination [3]. 

There has been interest in the implications of climate-related abiotic conditions and how these may affect the production and processing phases of tree nuts, especially pistachios and what impact such changes may have on infection by *A. flavus* and AFs contamination. However, very few, if any, studies have focused on addressing these issues. Most previous studies have focused on cereals, especially maize and wheat [6,7].

For example, the exposure of stored maize kernels with *A. flavus* to existing climate related conditions (30 °C, 400 ppm CO_2_, no drought stress) and to future climate-related abiotic factors (34 °C, 1000 ppm CO_2_, drought stress) showed that, while colonization was unaffected, AFB_1_ contamination was stimulated [6,7]. This was supported by increased expression of biosynthetic genes involved in AFs production, including an early structural gene (*aflD*) and a regulatory gene (*aflR*) in biosynthetic pathway. Indeed, a subsequent kinetic study also showed that this stimulation of AFB_1_ occurred when *A. flavus* was exposed on maize-based media to climate change scenarios [8]. 

A recent study of the water and temperature relationship of *A. flavus* strains isolated from pistachio nuts in Saudi Arabia showed that optimum growth on both pistachio nut-based media and raw pistachios was optimal at 35 °C in contrast to maize, where this is usually at 28–30 °C [3,9]. This certainly suggests that strains from pistachio nuts may have evolved to have better tolerance to high temperatures in production areas such as Iran and perhaps the USA from where the strains originated [3]. Indeed, optimal conditions for the growth of four strains of *A. flavus* from pistachios were found to be at 0.98–0.95 water activity (a_w_) and 30–35 °C. AFB_1_ production was also optimal at 30–35 °C and 0.98 a_w_. Interestingly, while less AFB_1_ was directly produced on raw shelled pistachio nuts the optimum and marginal boundary conditions were similar [3].

Drought stress interactions with elevated temperature and exposure to climate-related CO_2_ concentrations may be important. These factors were shown to influence and increased the colonization of maize cobs pre-harvest by *Fusarium verticillioides* and contamination with fumonisins [10]. Colonization of other commodities, such as stored green coffee by *Aspergillus westerdijkiae* observed optimal ochratoxin A (OTA) production under the existing climate-related condition of 400 ppm CO_2_ and intermediate drought stress/temperatures of 0.95–0.97 a_w_/30 °C. However, when coffee was stored and exposed to elevated CO_2_ (1000 ppm) and 35 °C, OTA production was stimulated, but only under increased water stress (0.95–0.90 a_w_) [11].

Interacting climate change related conditions will become more important over the next few decades and is important to understand the potential implications for food security and food safety. The actual estimated atmospheric CO_2_ concentration is, at present, approximately 417 ppm [12] and is predicted to double to 800 ppm (2×) or to 1000–1200 ppm (3×) in the future. This is suggestive of a temperature increase of 2–4 °C. Furthermore, rainfall patterns are also expected to change, with more extreme wet/dry episodes occurring [8,13]. Other important factors include the effect of light and also fluctuating day/night temperatures which can also impact colonization and toxin contamination of food commodities [14,15,16]. Previous studies have suggested that interactions between these critical abiotic factors may have a significant impact on fungal diseases of staple food crops and perhaps also on the contamination of food and feed with mycotoxins [17,18].

The objectives of this study were to evaluate the effects of three way interactions between climate change-related factors of temperature (35 vs. 37 °C) × water availability (0.98, 0.95, 0.93 a_w_) × exposure to CO_2_ (400 vs. 1000 ppm) on (a) growth, (b) AFB_1_ production, and (c) expression of biosynthetic genes (*aflR*, *aflD*) involved in AFs production in vitro on a pistachio nut-based medium for two strains of *A. flavus* isolated from pistachio nuts. This was complimented with similar studies in situ on stored raw pistachio nuts on growth and AFB_1_ contamination.

## 2. Results

### 2.1. In Vitro Effect of Climate Change Factors on Growth of Aspergillus flavus Strains on Pistachio Nut-Based Media

The effect of changing the three-way interacting climate-related abiotic factors on the growth of the two strains of *A. flavus* on the pistachio nut agar (PNA) nutritional media is shown in Figure 1. For both strains, growth on the pistachio-based medium was relatively similar with no significant differences in relation to the three interacting abiotic conditions. Colonization increased with more available water (0.98 a_w_) than under drier conditions at both 35 and 37 °C, regardless of CO_2_ exposure treatments. 

### 2.2. In Vitro Effect of Climate Change Interacting Factors on Aflatoxin B_1_ Production from Aspergillus flavus Strains on Pistachio-Based Media

Figure 2 shows the effect of the interaction between climate change-related abiotic factors on AFB_1_ production by the two strains of *A. flavus* (AB3 and AB10) examined on the milled pistachio-based media as the only nutritional source. Generally, AFB_1_ production was higher at 35 °C at the two CO_2_ levels (400 and 1000 ppm CO_2_) for both stains. At this temperature, the production was significantly increased at 1000 ppm CO_2_ and 0.98 a_w_ when compared with atmospheric air at 400 ppm CO_2_. At 37 °C, AFB_1_ production by the *A. flavus* strains when exposed to 1000 ppm CO_2_ was either decreased (strain AB3) or remained similar in the other strain.

### 2.3. In Vitro Effect of Climate Change Interacting Factors on Relative Genes Expression of the aflD and aflR Genes Involved in the Biosynthetic Pathway for Aflatoxin B_1_ Production

Figure 3 and Figure 4 show the effect of the climate change-related abiotic factors on the relative gene expression of the structural *aflD* and the regulatory gene *aflR* for the two strains. The control conditions (calibrator) used for comparisons between treatments was the 35 °C, 400 ppm CO_2_, 0.98 a_w_. For the *aflD* gene at 35 °C, the relative expression was higher at 400 ppm CO_2_ for both strains. However, for strain AB3, the expression was higher at 1000 ppm CO_2_ and 0.95 a_w._ However, at 37 °C, the expression was generally increased at 1000 ppm CO_2_ when compared with existing atmospheric CO_2_ levels. With regard to the regulatory *aflR* gene, in some a_w_ treatments the expression was higher in the 1000 ppm CO_2_ treatment at 37 °C for both strains. This suggests that the interaction between the three-climate change-related abiotic factors stimulated the expression of this gene which paralleled the effects on production of AFB_1_ under some conditions. 

### 2.4. In Situ Effect of Climate Change Factors on Growth and Aflatoxin B_1_ Production by Strains of Aspergillus flavus Strains on Pistachio Nuts

#### 2.4.1. Effects on Colonization of Raw Pistachio Nuts

The relative colonization rates of raw pistachio nuts suggest that regardless of temperature or CO_2_ exposure, there were practically no significant differences at 0.98 and 0.95 a_w_ treatments for both strains examined (Figure 5). Thus, exposure to 1000 ppm CO_2_ had little effect on colonization rates at both 35 and 37 °C suggesting good resilience to the imposed climate-related abiotic conditions. However, the growth of the AB3 strain was slightly decreased at 1000 ppm CO_2_ under the lowest a_w_ tested (0.93 a_w_) in the two incubation temperatures.

#### 2.4.2. In Situ Effect of Climate Change Interacting Factors on Aflatoxin B_1_ Production from *Aspergillus flavus* Strains on Pistachio Nuts

Figure 6 shows the effect of the three-way interacting climate-related abiotic factors on AFB_1_ production by the two strains of *A. flavus* after 10 days colonization of the layers of shelled raw pistachio nuts. There was an increase in AFB_1_ production when they were exposed to 1000 ppm CO_2_ for both strains, especially at 35 °C and 0.98 a_w_ (strain AB3) and 0.95 a_w_ (strain AB10). Overall, at 37 °C, there was significantly less AFB_1_ produced, regardless of the interacting abiotic conditions. There was also some indication that when exposed to 1000 ppm CO_2_ and 37 °C more AFB_1_ was produced than at 400 ppm CO_2_ exposure and the same temperature for strain AB3.

## 3. Discussion

The study has examined, for the first time, the potential effects of interacting climate change-related abiotic factors and the relative resilience of strains of *A. flavus* in terms of growth and AFB_1_ production on pistachio-based matrices and on raw pistachio nuts. The in vitro studies also examined the effects on structural and regulatory genes involved in the biosynthesis of aflatoxin-related secondary metabolites. Overall, these studies showed that there were no significant differences in relative growth rates of the two *A. flavus* strains in relation to existing and future climate-related environmental stresses, both in vitro on the PNA medium or in situ when colonizing raw pistachio nuts. Interestingly, on the milled pistachio nut-based nutritional media AFB_1_ production were stimulated by exposure to 1000 ppm CO_2_ but only at 35 °C. Colonization of raw pistachio nuts resulted in some stimulation of AFB_1_ at 37 °C but only with exposure to existing CO_2_ concentrations.

The present study focused on 35 °C and 37 °C as the two temperatures to compare. This is because the water and temperature relations of four strains, including the two in this study, of *A. flavus* isolated from pistachio nuts under prevailing CO_2_ conditions (400 ppm), showed that colonization was optimal at 35 °C and 0.98 a_w_, regardless of whether the pistachio-based medium was modified with ionic (NaCl) or non-ionic (glycerol) solutes [3]. Thus, an increase of 2 °C was chosen to represent climate-related increases in the next 10–15 years. Previous studies with maize, wheat and coffee have all used +4 °C and double or triple the existing CO_2_ levels and in some cases drought stress [11,13,19,20,21].

Most of the previous work on the resilience of *A. flavus* strains to climate change-related abiotic factors has focused on those originating from maize [8,22]. On a conducive defined nutritional medium for toxin production, it was found that growth was unaffected by 650 or 800 ppm CO_2_ exposure at 35 °C under freely available or water stress conditions. In addition, both *aflD* and *aflR* gene expression and phenotypic AFB_1_ production were significantly stimulated when compared to 30 °C + 400 ppm CO_2_ and 0.93–0.98 a_w_ [16,22]. Subsequent studies on stored maize grain confirmed this stimulation, although the stimulation was not as profound as that observed in vitro [7,16]. Work on other mycotoxigenic fungi in grapes with strains of *A. carbonarius* examined fluctuating extreme day/night temperature cycles and elevated CO_2_ (1000 ppm) and found an overall stimulation of growth and OTA production [23]. This contrasted with studies on coffee which showed stimulatory effects on OTA production by *A. westerdijkiae* strains but not strains of *A. carbonarius* [20].

The results obtained for the two strains of *A. flavus* in terms of relative expression of mycotoxin-related biosynthesis were different. They were both molecularly identified and compared with a type strain of *A. flavus* originating from maize. It is thus difficult to explain the differences found between strains AB3 and AB10. The relative expression of both *aflD* and *aflR* observed could be due to the molecular analyses after 10 days incubation. The structural *aflD* gene is early in the biosynthetic pathway and usually expressed earlier than the *aflR*. The results for the regulatory gene thus may be of more interest [20]. Previous studies have suggested that the *aflD* gene, on maize-based matrices is expressed after 4–5 days [24,25]. However, the *aflR* gene expression appeared to parallel the AFB_1_ production by both *A. flavus* strains. This was especially so at 35 °C + 1000 ppm CO_2_ at 0.98 a_w_ when compared to the control. 

Medina at el. [22] found that the maximum relative expression of *aflD* gene was at 34 °C + 400 ppm CO_2_ with a decrease in expression with elevated CO_2_ and water stress. In contrast, *aflR* gene expression was found to significantly increase, only under drier conditions (0.92 a_w_ + 650 ppm CO_2_). In contrast, the expression of both *aflD* and *aflR* genes increased significantly at 37 °C under treatment conditions of 0.95/0.92 a_w_ and 650 and 1000 ppm CO_2_. These increases were associated with an increase in AFB_1_ production. More recent studies of the impact of two way a_w_ × temperature interactions and three-way climate-related interacting factors showed that there were significant changes in the transcriptome of *A. flavus* both in vitro and in stored maize. Secondary metabolite pathways, sugar transporters and other related gene clusters were found to significantly change in RNAseq studies [7,16]. In the studies performed with *A. carbonarius* on grape-based media, fluctuating day/night temperatures and 1000 ppm CO_2_ resulted in an up-regulation of both structural (*AcOTApks*, *AcOTAnrps*, *AcOTAhal, AcOTAp450*, *AcOTAbZIP*) and regulatory genes of the velvet complex (*laeA*/*veA*/*velB*, “velvet complex”) involved in OTA biosynthesis [23]. Studies with *Fusarium langsethiae,* a non-xerophilic species, which grows under cooler conditions on oats was also found to responded to three-way climate-related abiotic factors. These showed that the *Tri5* gene expression was reduced in all conditions except at elevated temperature, 30 °C, when compared to 25 °C, and exposure to 1000 ppm CO_2_ with a 5.3-fold significant increase in expression. Other biosynthetic genes (*Tri6*, *Tri16*) were upregulated, in elevated CO_2_ conditions. In addition, mycotoxin production was higher at 25 °C than at 30 °C in vitro. In stored oats, at 0.98 a_w_, elevated CO_2_ led to a significant increase (73-fold) in T2/HT-2 toxin, especially at 30 °C [26]. 

Overall, the effects of three-way interacting climate-related abiotic factors on mycotoxin production by different toxigenic species have found differential effects. Thus, colonization and ochratoxin A (OTA) production by strains of *A. westerdijkiae* and *A. carbonarius* on coffee-based media and in stored coffee showed some differences [11,19,20]. Akbar et al. [20] showed that for *A. westerdijkiae*, while growth was relative unaffected, OTA production was stimulated by climate change-related interacting factors, both in vitro and in situ. However, for *A. carbonarius* there was no effect on growth or OTA production. Thus, differential effects of these interacting factors may occur. Vaughan et al. [10] found effects on fungal biomass during infection of maize by *F. verticillioides* in ripening maize, but no effect on fumonisins production when exposed to 650 ppm CO_2_. However, when drought stress was included, there was a stimulation of fumonisins production in ripening maize cobs [24].

Another important factor may be the relative changes in pest damage. It has been suggested that pest reproduction rates increase significantly under climate-related scenarios and more damage to ripening crops may also influence toxin production and perhaps the ratio of related toxins [7,13]. In addition, the impact of acclimatisation needs to be addressed. Studies by Vary et al. [27] showed that growing *F. graminearum* for 20 generation in elevated CO_2_ resulted in higher infection of ripening wheat, with increased head blight symptoms. Studies of acclimatization of *A. flavus* strains from pistachios showed that after five generations of sub-culturing at 1000 ppm CO_2_, some strains produced more AFB_1_ than the same strains which were non- acclimatized [13,28]. This suggests that more definitive studies are necessary on the acclimatisation of fungal pathogens and their effect on crops under climate-related scenarios to better understand their resilience and infection rates of different economically important commodities [13]. 

In terms of impacts of climate change scenarios, more studies are necessary to examine whether *A. flavus* strains may switch to the production of other mycotoxins, such as cyclopiazonic acid, or whether there could be a change in the ration of these mycotoxins and other secondary metabolites. This type of information together with ecophysiological data on optimal and marginal conditions for growth and AFB_1_ production [3] could be beneficial in the development of accurate predictive models on the relative risks of AFB_1_ toxin contamination of pistachios under future climate-related abiotic conditions.

## 4. Conclusions

This study has shown that *A. flavus* strains are very resilient when exposed to three-way interacting climate change-related abiotic factors. These was no impact on growth and colonization of both pistachio nut-based media and raw pistachio nuts. However, there were some effects in vitro on the relative expression of a structural and regulatory genes *(aflD, aflR*) and on AFB_1_ production. Exposure to elevated atmospheric CO_2_ affected AFB_1_ production with an increase in toxin production at 35 °C + 1000 ppm CO_2_. Significantly less AFB_1_ was produced at 37 °C + 1000 ppm CO_2_, regardless of the a_w_ level examined, when compared to the control (400 ppm CO_2_). 

## 5. Materials and Methods

### 5.1. Aspergillus flavus Strains

Two strains of *A. flavus* isolated from pistachio nut samples were molecularly identified using two sets of primers (ITS 1/2, ITS 3/4, Appendix A). They were coded as AB3 and AB10 and identification was further confirmed by comparison with a characterized type strain of *A. flavus* (NRRL 3357) from the Agricultural Research Service Culture Collection (USA), isolated from maize grain (Appendix A). They were all aflatoxin B_1_ producers [3]. 

### 5.2. Preparation of In Vitro Growth Media and Inoculation

The medium used was a 3% milled pistachio nut agar (PNA). To prepare this medium, raw unsalted pistachio nut were milled to a powder in a homogeniser. The milled pistachio powder was then sieved to obtain a uniform size. Thirty grams (30 g) of the pistachio powder and 20 g technical agar (Thermo Fisher Scientific Oxoid Ltd, Basingstoke, Hampshire, UK) and 0.05 g chloramphenicol (antibacterial agent) was added to 1 L distilled water for the basal medium (0.99 a_w_). The a_w_ was modified by initially making up mixtures of water/glycerol solutions by adding 122.5, 245.0 and 355 g/1 L glycerol to water. These were shaken vigorously and then measured and added, in a similar manner to water, to the pistachio nut flour and agar. This gave the target water availability conditions of 0.93, 0.95 and 0.98, respectively. The treatments were then autoclaved at 121 °C for 15 min. After autoclaving, the PNA treatments were cooled, mixed thoroughly, poured into 9 cm sterile Petri plates (17.5–20 mL per plate) and allowed to completely cool and solidify. The a_w_ was measured using an Aqualab 4TE (Decagon Instruments) and found to be within 0.003 a_w_ of the target values. The media a_w_ treatments were enclosed in separate closed polyethylene bags and stored at 4 °C until use.

The PNA plates were subsequently equilibrated at 25 °C and then centrally inoculated with the strains of *A. flavus* (AB3, AB10). Inoculum consisted of a conidial spore suspension of each strain made from fresh 5–7 days old growing cultures on PNA at 25 °C. The culture surface was gently scraped with sterile loop and conidia were transferred into sterile 25 ml Universal glass tubes containing 10 ml sterile water + 0.1% Tween 80 solution (Tween 80, ACROS organics, Fisher Scientific, Loughborough, UK). The concentration of the spore suspension was determined using a haemocytometer (Olympus BX40 microscope, Microoptical Co.; slide Marienfeld superior, Germany; microscope glass cover slips, No. 3, 18 × 18 mm, Chance Propper Ltd, Smethwick, Warley, UK) and adjusted by dilution with sterile water to 10^6^ spores/mL. The treatments and replicates were centrally inoculated with 10 µL of the spore suspension. The different a_w_ treatments and replicates were then centrally inoculated with the individual *A. flavus* strains (AB3, AB10). Three replicates of each strain were placed in the CC environmental chambers and the Equilibrium Relative Humidity (ERH) was maintained with glycerol/water solutions (2 × 500 mL) in beakers to maintain the target a_w_ levels. The experiments were carried out twice.

### 5.3. Climate Change Experimental System

The environmental chambers used included an inlet and outlet valve on either side and could be sealed during exposure and storage [8]. The in vitro and in situ treatments and replicates were placed into individual separate chambers. They were flushed with either air (400 ppm) or 1000 ppm CO_2_ (speciality gases; British Oxygen Company, Guildford, Surrey, UK). The chambers were flushed with the required CO_2_ concentrations at a rate of 2 L/min to replace 3× chamber volume. This process was repeated every two days immediately after growth measurements were recorded and the chambers then sealed again. The treatments were incubated at 35 and 37 °C. Trial measurements showed that the CO_2_ build up during incubation, because of the chamber volume was only increased slightly over the two days period by between 125–150 ppm when measured using Gas Chromatography.

### 5.4. Effect of Interacting Climate Change Abiotic Factors on In Situ Colonisation of Pistachio Nuts by Aspergillus flavus

#### 5.4.1. Moisture Adsorption Curve

The pistachio nuts used in this study were gamma irradiated at 12–15 kGys (Synergy Health Sterilisation UK Ltd., Swindon, Wiltshire, UK) to remove any resident microbiota present. To accurately modify the a_w_ of the raw pistachio nuts, a water adsorption curve was developed. The relationship between added water and a_w_ was obtained by adding known amounts of water to 5 g sub-samples of raw pistachio nuts in 25 mL Universal glass bottles. These were shaken, sealed and left at 4 °C overnight. After thorough mixing and equilibration at 25 °C, the a_w_ was determined for each sub-sample using the Aqualab 4TE water activity meter (Aqualab 4TE; Decagon Devices, Inc., Pullman, WA, USA). The data for a_w_ vs. amounts of added water were plotted. This was used to accurately add known quantities of sterile water to the pistachio nuts to accurately obtain the target a_w_ treatment values. We used the following modified a_w_ levels of the raw pistachio nuts: 0.93, 0.95 and 0.98 (=13–14, 18–19, 26–27% moisture content, m.c.). The pistachio nuts were placed in sterile Duran flasks with the added sterile water and equilibrated at 4 °C overnight with thorough periodic mixing. The pistachio nuts were equilibrated in the laboratory and the a_w_ values confirmed.

#### 5.4.2. Inoculation and Growth Assessment of Effects of Climate Change-Related Abiotic Factors on Colonization of Pistachio Nuts

Single layers of treatment pistachio nuts were spread into 9 cm Petri plates in a sterile flow bench. These were centrally inoculated with an individual *A. flavus* strain (AB3, AB10). Inoculum was obtained by spread plating 0.2 mL of a 10^6^ mL conidial suspension onto 9 cm PNA Petri plates. These were incubated at 25 °C overnight. A 4 mm diameter surface-sterilised cork-borer was used to obtain agar discs containing germlings to centrally inoculate the layers of raw pistachio nut treatments using a surface-sterilised inoculum needle. The experiment was carried out with three replicates per treatment at 35 and 37 °C and repeated once. Inoculated treatment and replicate Petri plates for each strain were immediately placed into the environmental chambers, closed and flushed with CO_2_ as described previously. The colonization rates were measured in two directions at right angles to each other, every two days for ten days.

### 5.5. Gene Expression Studies

Sampling was carried out following ten days incubation in triplicate for gene expression studies. The biomass was aseptically harvested, immediately frozen in liquid nitrogen, and stored at −80 °C for subsequent RNA extraction.

#### 5.5.1. RNA Isolation from Pistachio Nuts

Using the bead-beating method published by Leite et al. [29], RNA was extracted with some modifications. Frozen biomass (150 mg) was transferred into an autoclaved 2 mL extraction tube containing 0.5 mm sized glass beads. Then, 1 mL of RLT buffer (provided by the RNeasy^®^ Plant Mini Kit) (Qiagen, Hilden, Germany) supplemented with 10 µL of β-mercaptoethanol was added. The tubes were immediately frozen in liquid nitrogen. After a quick vortex to help disrupt the mycelium, samples then were placed on ice for thawing. The tubes were agitated for 25 s followed by a 5 s interval and another 25 s of agitation. This mixture was centrifuged at 9820 g for 5 min at 4 °C in a temperature-controlled centrifuge system. A pre-frozen 2 mL Safe-Lock tube (Eppendorf, Hamburg, Germany) was used to place the collected supernatant. According to instructions of the RNeasy^®^ Plant Mini Kit (Qiagen, Germany), the RNA purification was carried out. RNA obtained was eluted in 50 µL of RNase free water and kept at −80 °C until used for reverse transcription. The RNA concentration and purity (A260/A280 ratio) were determined spectrophotometrically using a 2.5 µL aliquot on the Picodrop^TM^ (Spectra Services Inc., Phoenix, AZ, USA).

#### 5.5.2. Primers and Probes

Nucleotide sequences of primers and probes used in this study are included in Table 1 [18]. The design of the primer pairs of nortaq-1/nortaq2 and aflRtaq1/aflRTaq2 and the hydrolysis probe norprobe and *AflR*probe were, respectively, designed based on the *aflD* and *aflR* genes involved in the aflatoxin biosynthetic pathway. The primer pair bentaq1/bentaq2 and the hydrolysis probe benprobe were designed and relied on the β-tubulin gene. The norprobe and *AflR*probe were labelled at the 5′ end with the reporter molecule 6-carboxyfluorescein (FAM) and at the 3′ end with the quencher Black Hole Quencher 2 (BHQ2). However, the reporter cyanine-5 (CY5) was used for the benprobe and labelled at the 5′ end with the quencher BHQ2 at the 3′ end. 

#### 5.5.3. Reverse Transcription to Convert cDNA from mRNA

Five μL of total RNA (≈500 ng) was used to synthesise cDNA from mRNA. For this, the Omniscript RT kit (Qiagen) was used, and the protocol described by the manufacturer was followed. The reverse transcription component mix consisted of 2 µL 10× Buffer RT, 2 µL dNTP Mix (5 mM each dNTP), 2 µL Oligo-dT primer (10 µM), 1 µL RNase inhibitor (10 units/µL), 1 µL Omniscript Reverse Transcriptase, 7 µL RNase-free water, and 5 µL total RNA. Once the mix was ready, it was incubated for 60 min at 37 °C. cDNA was stored at −20 ℃ for long-term storage.

#### 5.5.4. Amplification of *aflD* and *aflR* Genes Through Real-Time Quantitative PCR (RT-qPCR)

To amplify the structural gene *aflD (nor-1)* and the regulatory gene *aflR* of the aflatoxin biosynthetic pathway as the target genes, a quantitative RT-qPCR assay was used. The β-tubulin gene was used as a control gene [9,18]. The Bio-Rad CFX96 Real Time PCR Detection System (Bio-Rad, Watford, UK) was used to perform two RT-qPCR assays to amplify the *aflD* gene and the housekeeping β-tubulin gene in the first one, and the other one to quantify the *aflR* gene expression using the β-tubulin gene as control [16]. They were prepared in triplicates of 12.5μL reaction mixture in MicroAmp optical 96-well reaction plates and sealed with optical adhesive covers (Bio-Rad). Three replicates of an RNA control sample together with a template-free negative control were also included in the runs. The TaqMan system with primers and probes were used in all cases. The reaction mixtures consisted of 6.25 μL Premix Ex TaqTM (Takara Bio Inc., Otsu, Shiga, Japan), 830 nM of each primer, 330 nM of each probe, and 1.5 μL of cDNA template in a final volume of 12.5 μL. The optimal thermal cycling conditions included an initial step of 10 min at 95 °C and all 45 cycles at 95 °C for 15 s, 55 °C for 20 s and 72 °C for 30 s. *C*_t_ determinations were automatically performed by the instrument using default parameters and obtained from the BIO-RAD detection system.

Data analysis was carried out using the software CFX Manager^TM^ Software (Bio-Rad). Relative quantification of the expression of *aflD* and *aflR* genes were carried out using the housekeeping gene β-tubulin as an endogenous control to normalise the quantification of the mRNA target for differences in the amount of total cDNA added to the reaction in the relative quantification assays and used for all treatments. The expression ratio was calculated as previously described by Livak and Schmittgen [30]. Prior to the analysis, we found that the experimental treatments did not influence the expression of the internal control gene, and the amplification efficiencies of the target and reference genes were practically equal (93.1% for *aflR* and 95.2% for β-tubulin genes). This method allows for the calculation of the expression ratio of a target gene between a tested sample and its relative calibrator (“control” sample). In this work, the calibrator corresponded to *A. flavus* strains grown at 35 °C and 0.98 a_w_ at atmospheric air (400 ppm CO_2_). Log_2_ values of the relative expression of the *aflD* and *aflR* genes were graphically represented. The statistical design was factorial CRD, 2 factors and the statistical analysis obtained using SPSS® software.

### 5.6. Quantification of Aflatoxin B_1_ Production

Aflatoxin B_1_ quantification: Preparation of aflatoxin standards: A 200 μL stock solution of aflatoxins (B_1_, B_2_, G_1_, G_2_) standard in methanol containing 250 ng AFB_1_ was prepared and pipetted into 2 mL Eppendorf tubes for overnight evaporation until dryness in a fume hood similar to the samples.

In vitro aflatoxin B_1_ analyses: Colony Extraction: Initially agar plugs were cut out across the diameter of colonies using a surface sterilised 4 mm diameter cork borer (approx. 4–6). The agar plugs were placed in pre-weighed 2 mL Eppendorf tube and weighed again. Five-hundred millilitres of HPLC-grade chloroform was added to the tubes and shaken for 30 min using a KS 501 digital orbital shaker (125 rpm; IKA (R) Werke GmbH & Co. KG, Germany). The chloroform extract was transferred to a new Eppendorf tube, dried gently under air for derivatisation.

Derivatisation of aflatoxin B_1_ extract: Derivatisation of the AFB_1_ extract was performed according to the AOC method (Kok, 1994). First, 200 μL hexane was added to the tube followed by 50 μL of triflouroacetic acid. The mixture was vortexed for 30 s and left for 5 min. A mixture of water:acetonitrile (9:1) was then added to the tube, and vortexed for 30 s and left for 10 min to allow for separation of the layers. Then, the aqueous layer was filtered using a syringe nylon filter (13 mm × 0.22 μm; Jaytee Biosciences Ltd., Herne Bay, UK) into amber salinized 2 mL HPLC vials (Agilent, Santa Clara, CA, USA) before HPLC analysis. All analytical reagents used were HPLC-grade.

Quantification of aflatoxin B_1_ with High Performance Liquid Chromatography HPLC: A reverse-phase HPLC with fluorescence detection was used to confirm the identity and quantify AFB_1_. An Agilent 1200 series HPLC system was used for the analysis. It consisted of an in-line degasser, auto sampler, binary pump and a fluorescence detector (excitation and emission wavelengths of 360 and 440 nm, respectively). Separation was achieved using a C18 column (Phenomenex Gemini; 150 × 4.6, 3 μm particle size; Phenomenex, Torrance, CA, USA) with a Phenomenex Gemini C18 3 mm, 3 μm guard cartridge. Isocratic elution with methanol:water:acetonitrile (30:60:10, *v*/*v*/*v*) as the mobile phase was performed at a flow rate of 1.0 mL/min. The injection volume was 20 µL. A set of standards was injected (1 to 5 ng AFB_1_, AFB_2_, AFG_1_ and AFG_2_ per injection) and standard curves were generated by plotting the area underneath the peaks against the amounts of AFB_1_ standard injected.

#### Isolation and Quantification of Aflatoxin B_1_ in Pistachio Nuts

The pistachio nut samples were all dried in a drying oven at 50 °C in the dark. They were subsequently ground (Waring blender, Merck Ltd., Feltham, UK) and weighed (25 g). The background aflatoxin B_1_ levels in the nuts used in the experiments was 0.015 ng/g. This was taken into account as a correction factor in the final quantification of the results. Acetonitrile/water 60/40 (100 mL) was used as an extraction solvent. The mixture was blended for 3 min and the extract filtered into a smaller sample container. PBS buffer (pH 7.4, Thermo Fisher Scientific) was used for sample dilution, then the diluted extract was passed through an Immunoaffinity Column (IAC; AflaStar™; Romer Labs, Tulln an der Donau, Austria) with a flow rate between 1–3 mL/min. The column was rinsed with 2 × 10 mL sterile distilled water. HPLC-grade methanol (1.5–3 mL) was then applied to the column and the eluent was collected in a new amber glass vial and left to dry overnight at room temperature before the derivatisation step as detailed previously.

### 5.7. Statistical Analysis

Three replicates per treatment were used in all experimental studies and carried out twice. Means were obtained by taking the average of each of the three measurements with the standard error of the means (±SE). Datasets were tested for normality and homoscedasticity using the Shapiro–Wilk and Levene test, respectively. Analysis of Variance (ANOVA) was applied to analyse the variation of means with 95% confidence interval. Normal distribution of data were checked by the normality test Kolmogorov–Smirnov using Minitab statistical software. Fisher’s Least Significant Difference (LSD) was used to identify differences between the means with *p* ≤ 0.05 as significant difference using the same statistical software.

## Figures and Tables

**Figure 1 toxins-13-00385-f001:**
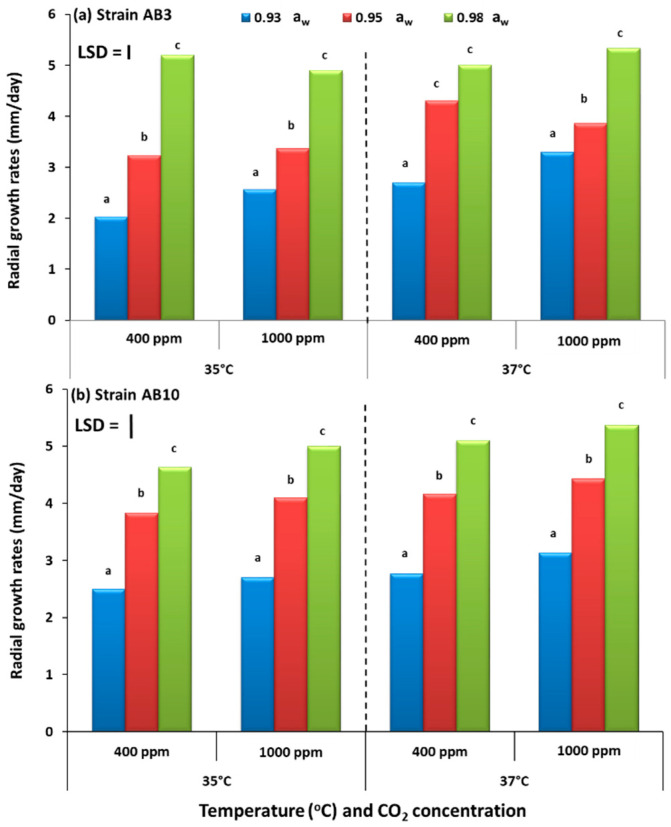
Comparison of growth rates of (mm/day) of *A. flavus* strain AB3 (**a**) and strain AB10 (**b**) grown on pistachio nut-based media and incubated at 35 and 37 °C under different concentrations of CO_2_ (400 vs. 1000 ppm) at 0.93–0.98 water activity (a_w_). Bars indicate Fisher’s Least Significant Difference (LSD, *p* ≤ 0.05). Different letters indicate significant differences.

**Figure 2 toxins-13-00385-f002:**
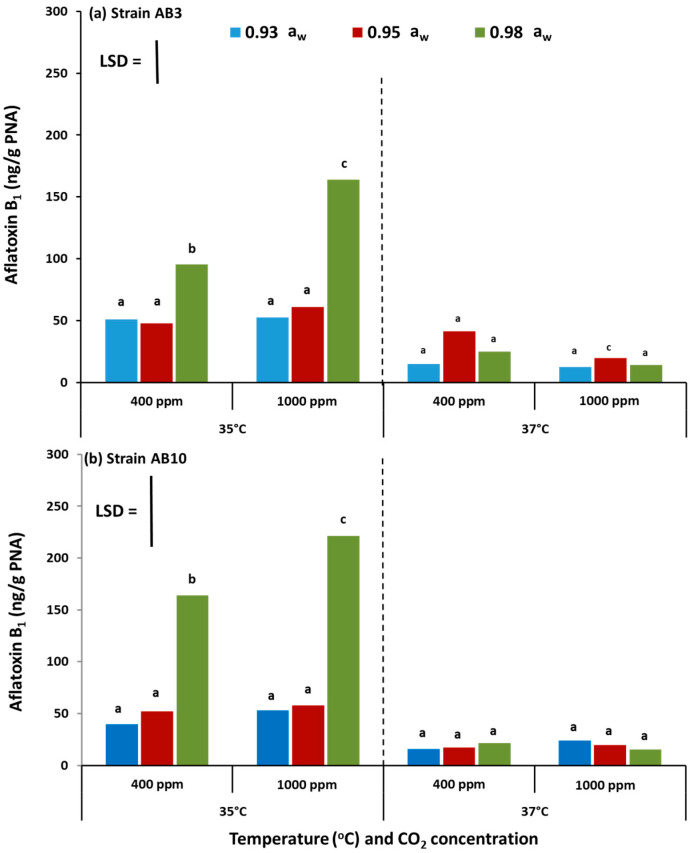
Impact of climate change-related interacting abiotic factors on aflatoxin B_1_ production by *A. flavus* strain AB3 (**a**) and strain AB10 (**b**) grown on pistachio nut-based media for a period of 10 days. Bars indicate Fisher’s Least Significant Difference (LSD, *p* ≤ 0.05). Different letters indicate significant differences.

**Figure 3 toxins-13-00385-f003:**
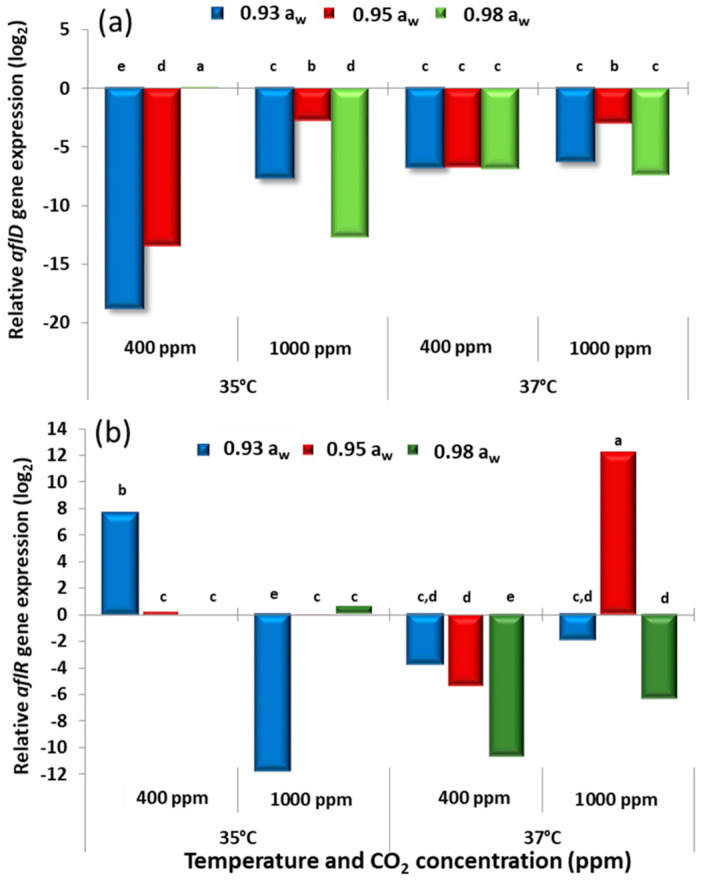
Effect of climate change-related interacting abiotic factors on relative expression of (**a**) *aflD* structural gene and (**b**) *aflR* regulatory gene of strain AB3 of *A. flavus* after 10 days incubation on a milled pistachio nut agar. The calibrator corresponded to *A. flavus* strains grown at 35 °C and 0.98 a_w_ in atmospheric air (400 ppm CO_2_), and has a value equal to 0. Different letters indicate significant differences (*p* ≤ 0.05).

**Figure 4 toxins-13-00385-f004:**
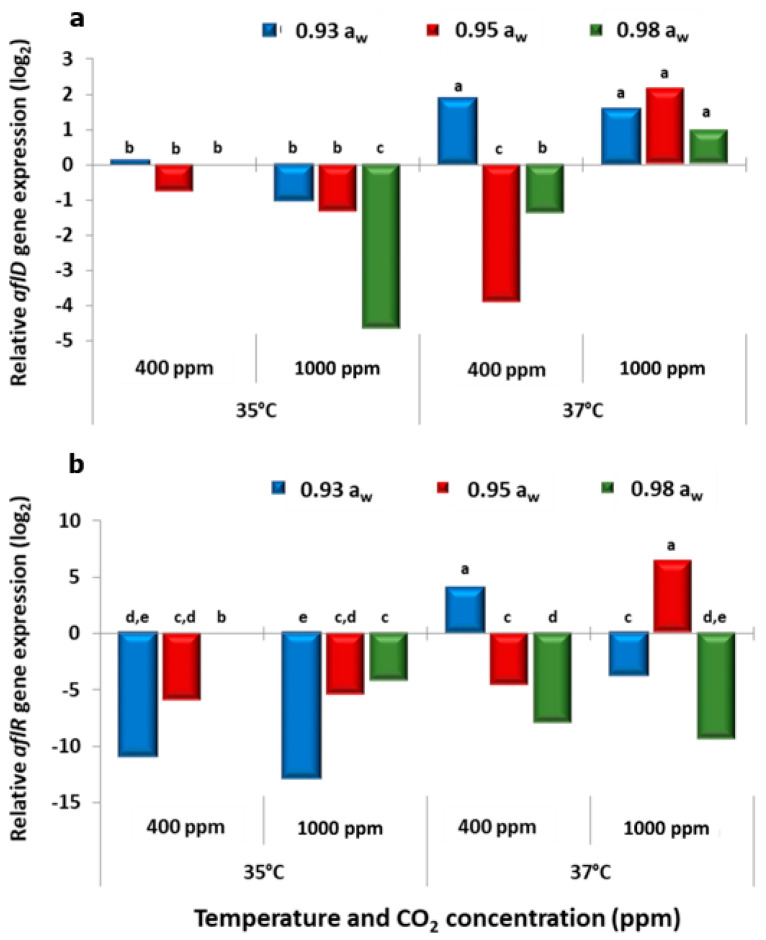
Effect of climate change-related interacting abiotic factors on relative expression of (**a**) *aflD* structural gene and (**b**) *aflR* regulatory gene of strain AB10 of *A. flavus* after 10 days incubation on a milled pistachio nut agar. The calibrator corresponded to *A. flavus* strains grown at 35 °C and 0.98 a_w_ in atmospheric air (400 ppm CO_2_) and has a value equal to 0. Different letters indicate significant differences (*p* ≤ 0.05).

**Figure 5 toxins-13-00385-f005:**
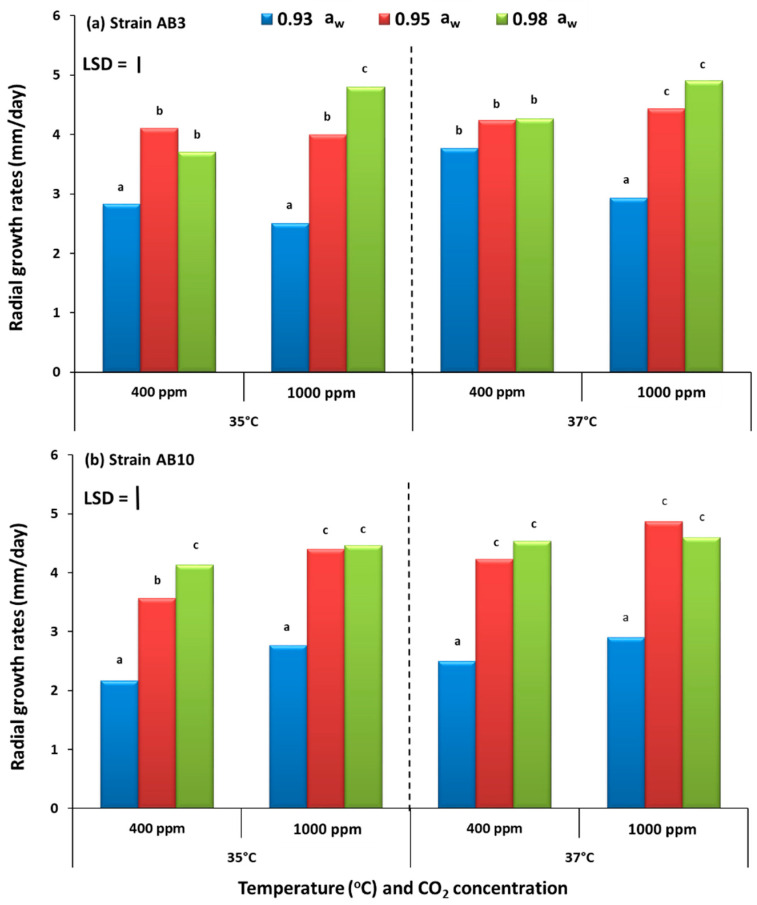
Effect of the three-way interacting climate-related abiotic factors on colonisation rates (mm/day) of single layers of raw pistachio nuts by *A. flavus* strain AB3 (**a**) and strain AB10 (**b**) over a period of 10 days. Bars indicate Fisher’s Least Significant Difference (LSD, *p* ≤ 0.05). Different letters indicate significant differences.

**Figure 6 toxins-13-00385-f006:**
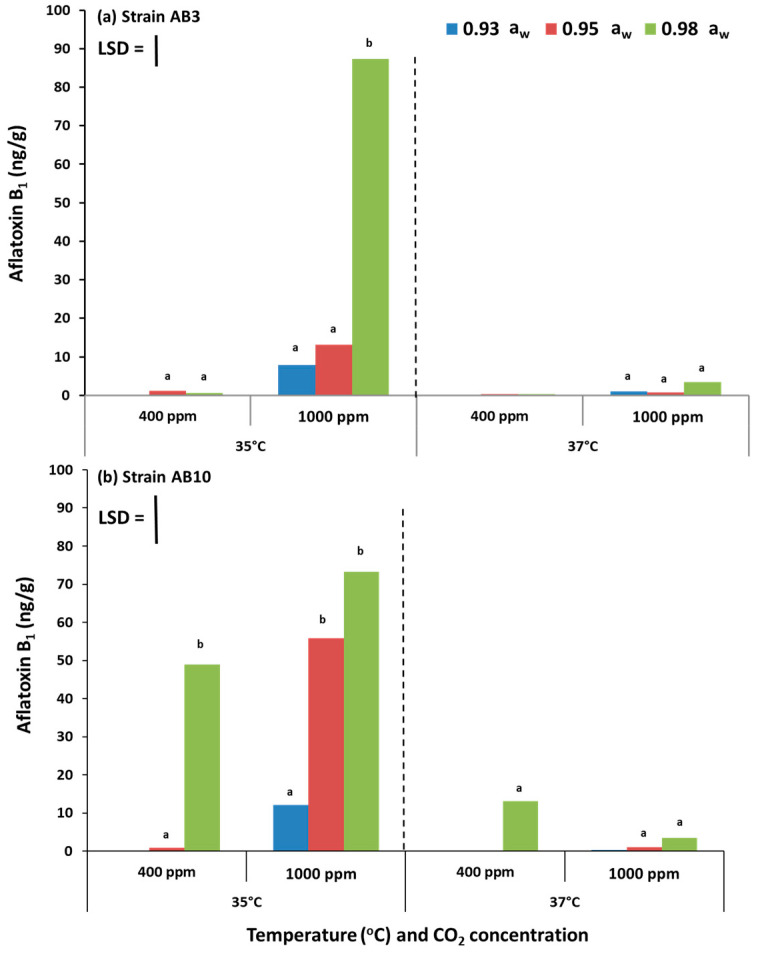
Comparison of the aflatoxin B_1_ production by *A. flavus* strain AB3 (**a**) and strain AB10 (**b**) when grown on single layers of raw shelled pistachio nuts and incubated under interacting climate-related abiotic factors. Bars indicate Fisher’s Least Significant Difference (LSD, *p* ≤ 0.05). Different letters indicate significant differences.

**Table 1 toxins-13-00385-t001:** Nucleotide sequences of primers for RT-qPCR assay designed on the basis of *AflD (nor-1), AflR* and β-tubulin genes.

Gene	Primers and Probes	Primer Sequence	Position
*AflD*	nortaq1	GTCCAAGCAACAGGCCAAGT	516 ^a^
	nortaq2	TCGTGCATGTTGGTGATGGT	562 ^a^
	norprobe	[FAM]TGTCTTGATCGCGCCCG[BHQ2]	537 ^a^
*AflR*	aflRTaq1	TCGTCCTTATCGTTCTCAAGG	1.646 ^b^
	aflRTaq2	ACTGTTGCTACAGCTGCCACT	1.735 ^b^
	aflRprobe	[FAM]AGCAGGCACCCAGTGTACCTCAAC[BHQ2]	1.6889 ^b^
β-tubulin	Bentaq1	CTTGTTGACCAGGTTGTCCAT	65 ^c^
	Bentaq2	GTCGCAGCCCTCAGCCT	99 ^c^
	benprobe	[CY5]CGATGTTGTCCGTCGCGAGGCT[BHQ2]	82 ^c^

^a^ Positions are in accordance with the published sequence of the *aflD* gene of *Aspergillus flavus* (GeneBank accession no. XM_002379908.1). ^b^ Positions are in accordance with the published sequence of *aflR* gene of *Aspergillus flavus* (GeneBank accession no. AF441435.2). ^c^ Positions are in accordance with the published sequence of ß-tubulin gene of *Aspergillus flavus* (GeneBank accession no. AF036803.1).

## Data Availability

The raw data supporting the Results and Conclusions of this article are deposited and available via the corresponding author at Cranfield University.

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
