# Peer review of "Impacts of Climate Change Interacting Abiotic Factors on Growth, aflD and aflR Gene Expression and Aflatoxin B1 Production by Aspergillus flavus Strains In Vitro and on Pistachio Nuts"

_toxins, 2021, doi:10.3390/toxins13060385_

Round 1

Reviewer 1 Report

The present manuscript entitled ‘Impacts of climate change interacting abiotic factors on growth, 2 aflD and aflR gene expression and aflatoxin B1 production by 3 Aspergillus flavus strains in vitro and on pistachio nuts’ present data in the possible impacts of climate change on the growth of Aspergillus flavus and aflatoxin contamination. The data is interesting and show that aflatoxin might be affected on both ways with climate change with an increase with higher CO2 at low temperatures and a decrease at higher temperatures. The experiment seems to be carried out well and the data looks sound to be published in Toxins. However, there are some issues that need to be corrected before publication.

First, the grammar needs to be improved all throughout the manuscript. Some sentences miss the whole meaning and can’t be understood. Also, paragraphs have mixing ideas and at the end, the idea the authors were trying to state are not well understood. Definitively, Introduction needs to be rewritten and organize to clearly state the main idea, the hypothesis and the objectives of the research. In Results as presented the authors analyzed treatments as a group, not by factors. It is difficult to really determine what of the factors (temperature, water or CO) are the ones having the effects. They need to present a factorial analysis to really determine what factors or interactions are relevant in determine both the fungi growth and aflatoxin contamination. Perhaps, presenting a table with the statistical analysis differentiating between 35 and 37 C, among 0.93, 0.95 and 0.98 Aw, and between 400 and 1000 ppm CO will help. Also, the discussion as written  needs to connect the significance of the studies cited to the present study. I don’t see that. They need to state what is the relevance of the studies in other fungi and mycotoxins in relation to Aspergillus flavus and aflatoxin contamination and how the interaction of the factors tested affect both the fungus growth and aflatoxin contamination. The way it is written looks more like a literature review than a discussion. Conclusions section should give direct statements of what was found or proved in the experiments. The last part of the conclusion section on the impacts of climate change, although important, it is quite speculative and not a direct statement based on the results of the experiments. It should be moved to the Discussion section as a recommendation.  Materials and Methods section look sound and I don’t think it needs any improvement. However, there is no evidence that the experiment was repeated (at least twice) to confirm and validate the results. Drawing conclusion with an experiment performed only one time and with very few replicates (only 3), especially for aflatoxin analysis is risky. If the experiment was repeated they need to say so and present the data for the two experiments or if they obtained similar data in both experiments they need to show the evidence. They need to show evidence of the reproducibility of the results. 

Some specific comments and suggestions:

Aspergillus flavus is a scientific name and has to be in italics. There are several instances where it is not and need to be consistent.

There are many spelling mistakes (typos) throughout the manuscript (e.g.  P1L9 atnospheric, P1L13-14 subsequnetly , P1L17 stidies) that need to be corrected.

In the abstract the authors are consistently stating that treatments were higher than the others, but they don’t give the magnitude of the differences and/or if they were significantly. They need to give the specific data.

There are few instances where “especially” is used twice in the same sentence. It doesn’t read well.

Colonization (American English) and colonisation (British English) are both used interchangeable in the manuscript. Please use consistently either spelling, but not both. It could be confusing. 

Need to improve grammar. They are mixing present tense with future tense in the same sentence (P2 L81-82). It is confusing. 

Author Response

All changes are highlighted in blue text.

The present manuscript entitled ‘Impacts of climate change interacting abiotic factors on growth, 2 aflD and aflR gene expression and aflatoxin B1 production by 3 Aspergillus flavus strains in vitro and on pistachio nuts’ present data in the possible impacts of climate change on the growth of Aspergillus flavus and aflatoxin contamination. The data is interesting and show that aflatoxin might be affected on both ways with climate change with an increase with higher CO2 at low temperatures and a decrease at higher temperatures. The experiment seems to be carried out well and the data looks sound to be published in Toxins. However, there are some issues that need to be corrected before publication.

First, the grammar needs to be improved all throughout the manuscript. Some sentences miss the whole meaning and can’t be understood.

Also, paragraphs have mixing ideas and at the end, the idea the authors were trying to state are not well understood.

Definitively, Introduction needs to be rewritten and organize to clearly state the main idea, the hypothesis and the objectives of the research. In Results as presented the authors analyzed treatments as a group, not by factors. It is difficult to really determine what of the factors (temperature, water or CO) are the ones having the effects. They need to present a factorial analysis to really determine what factors or interactions are relevant in determine both the fungi growth and aflatoxin contamination. Perhaps, presenting a table with the statistical analysis differentiating between 35 and 37 C, among 0.93, 0.95 and 0.98 Aw, and between 400 and 1000 ppm CO will help. Also, the discussion as written  needs to connect the significance of the studies cited to the present study. I don’t see that. They need to state what is the relevance of the studies in other fungi and mycotoxins in relation to Aspergillus flavus and aflatoxin contamination and how the interaction of the factors tested affect both the fungus growth and aflatoxin contamination. The way it is written looks more like a literature review than a discussion. Conclusions section should give direct statements of what was found or proved in the experiments. The last part of the conclusion section on the impacts of climate change, although important, it is quite speculative and not a direct statement based on the results of the experiments. It should be moved to the Discussion section as a recommendation.  Materials and Methods section look sound and I don’t think it needs any improvement. However, there is no evidence that the experiment was repeated (at least twice) to confirm and validate the results. Drawing conclusion with an experiment performed only one time and with very few replicates (only 3), especially for aflatoxin analysis is risky. If the experiment was repeated they need to say so and present the data for the two experiments or if they obtained similar data in both experiments they need to show the evidence. They need to show evidence of the reproducibility of the results. 

Answer: We have made all the minor changes recommended. We have separated where possible the paragraphs with separate ideas in the Introduction. Although we to some extent disagree about the Discussion, we have made changes to try and improve this. The Conclusions have also been changed.

The experiments were all carried out twice with similar results. In terms of the different factors, we have recently shown the effect of aw and temperature on both growth and AFB1 production by 4 different strains including the two used in this study. Previous studies have shown that it is the interaction between the three factors which can have an impact as opposed to individual or two-way interacting factors (Medina et al., 2020; Baazeem et al., 2021). Especially related to toxin production.

Some specific comments and suggestions:

Aspergillus flavus is a scientific name and has to be in italics. There are several instances where it is not and need to be consistent.

There are many spelling mistakes (typos) throughout the manuscript (e.g.  P1L9 atnospheric, P1L13-14 subsequnetly , P1L17 stidies) that need to be corrected.

Answer: All now corrected

In the abstract the authors are consistently stating that treatments were higher than the others, but they don’t give the magnitude of the differences and/or if they were significantly. They need to give the specific data.

Answer: modified and statistical significance mentioned.

There are few instances where “especially” is used twice in the same sentence. It doesn’t read well.

Answer: changed

Colonization (American English) and colonisation (British English) are both used interchangeable in the manuscript. Please use consistently either spelling, but not both. It could be confusing. 

Answer: changed to make this consistent

Need to improve grammar. They are mixing present tense with future tense in the same sentence (P2 L81-82). It is confusing. 

Answer: Hopefully now improved and better

Reviewer 2 Report

toxins-1178230-peer-review-v1

Reviewer summary.

The goal of this work is to investigate and characterize the effect of temperature, water availability, and CO2 exposure (authors call this climate change-related three way interactions) on fungal growth and colonization, AFB1 production, and expression aflR and aflD in two representative A. flavus strains, AB3 and AB10 in vitro and in-situ using pistachio nuts as a substrate. Major findings were that three-way interaction of abiotic factors did not affect growth and colonization of AB3 and AB10 on PNA and pistachio nuts respectively (authors concluded that strains were resilient to factors tested); AFB1 levels were elevated at 35C but mostly not at 37C on PNA. The observation of AFB1 production in Figure 2 is predictable, however, differences in gene expression levels in Figure 3 and 4 from AB3 and AB10 are difficult to interpret (not sure what do the data really mean?; differences between gene expression levels of two strains are also interesting). Authors found that in situ studies on pistachio nuts generally follow observations in vitro on PNA relative to growth expressed by colonization rates, although the magnitude of AFB1 production is quite different between treatments. Outcomes from this work will highlight the importance of A. flavus on pistachio nuts and the impact of climate-change related mycotoxin contamination on food and agricultural products. I have highlighted some strengths in this study as well as some minor comments and suggestions.

Strengths.

  • Authors demonstrated clear goals and motivation to conduct study
  • Strength in multifactorial three-way interaction (temp, water, CO2) as treatments, phenotypic characterization (growth, AFB1 levels, aflD and aflR gene expression), and use of two representative strains, AB3 and AB10, use of in vitro PNA and in situ pistachio nuts
  • Also appreciated the authors including a standard in vitro laboratory treatment condition to compare with on pistachio nuts
  • The authors did a really nice job on the experimental design and research questions
  • The authors did an excellent job with gene expression approach, re: the use of qPCR, use of internal control (b-tubulin; reported that treatments did not influence expression of internal control gene) and use of the expression ratio of a target gene between tested sample and its relative calibrator
  • Overall, he authors also did a really nice job in the discussion highlighting the rationale for each finding and providing supporting observations from other studies, especially providing perspective on next steps and future studies – especially related to acclimatization of A. flavus strains over generations and the potential impact on AF production.

Minor comments throughout text.

Abstract.

L6 fungi (spelling, sp)

L15 pistachio (sp)

L17 studies (sp)

Introduction.

L108 and L126 section 2.2. is repeated twice

Not sure if authors had previously hypothesized on outcomes of research questions – might be worthwhile including hypothesis for this study

Results.

Figure 3 and 4- based on differences in expression pattern of aflR and aflD across climate-related abiotic factors, it is clear that there are genetic differences in AB3 and AB10 – but you don’t see significant chances in AFB1 levels between both strains – explanation?

Figure 3. The difference in expression of aflR and aflD is interesting (I wonder whether the difference is due to time of RNA extraction, 10 days); typically wouldn’t you expect expression of nor-1 and aflR be in the same direction? See comment on time dependent expression L363; authors should discuss this to help explain gene expression data

Discussion. N/A

Materials and Methods.

L296 – I do not see supp. Table 1

L313 – how genetically similar/different AB3 v. AB10 is?

L338 -  does gamma radiation at 12-15 kGys impact fungi?

L363 – ten day incubation – is this optimal time for aflR and aflD gene expression levels?  In standard rich liquid media (for example, YES) we know that aflR peaks between 48-72 hours, do authors know what is the time dependent gene expression level in PNA? I believe this is important for the interpretation of gene expression data.

L384 – I would suggest that authors keep these genes consistent, perhaps Nortaq-1/nortaq2 (aflD)? OK, I see this in Table 1.

Author Response

All changes highlighted in blue.

Reviewer summary.

The goal of this work is to investigate and characterize the effect of temperature, water availability, and CO2 exposure (authors call this climate change-related three way interactions) on fungal growth and colonization, AFB1 production, and expression aflR and aflD in two representative A. flavus strains, AB3 and AB10 in vitro and in-situ using pistachio nuts as a substrate. Major findings were that three-way interaction of abiotic factors did not affect growth and colonization of AB3 and AB10 on PNA and pistachio nuts respectively (authors concluded that strains were resilient to factors tested); AFB1 levels were elevated at 35C but mostly not at 37C on PNA. The observation of AFB1 production in Figure 2 is predictable, however, differences in gene expression levels in Figure 3 and 4 from AB3 and AB10 are difficult to interpret (not sure what do the data really mean?; differences between gene expression levels of two strains are also interesting). Authors found that in situ studies on pistachio nuts generally follow observations in vitro on PNA relative to growth expressed by colonization rates, although the magnitude of AFB1 production is quite different between treatments. Outcomes from this work will highlight the importance of A. flavus on pistachio nuts and the impact of climate-change related mycotoxin contamination on food and agricultural products. I have highlighted some strengths in this study as well as some minor comments and suggestions.

Strengths.

  • Authors demonstrated clear goals and motivation to conduct study
  • Strength in multifactorial three-way interaction (temp, water, CO2) as treatments, phenotypic characterization (growth, AFB1 levels, aflDand aflR gene expression), and use of two representative strains, AB3 and AB10, use of in vitro PNA and in situ pistachio nuts
  • Also appreciated the authors including a standard in vitrolaboratory treatment condition to compare with on pistachio nuts
  • The authors did a really nice job on the experimental design and research questions
  • The authors did an excellent job with gene expression approach, re: the use of qPCR, use of internal control (b-tubulin; reported that treatments did not influence expression of internal control gene) and use of the expression ratio of a target gene between tested sample and its relative calibrator
  • Overall, he authors also did a really nice job in the discussion highlighting the rationale for each finding and providing supporting observations from other studies, especially providing perspective on next steps and future studies – especially related to acclimatization of A. flavus strains over generations and the potential impact on AF production.

Minor comments throughout text.

Abstract.

L6 fungi (spelling, sp)

L15 pistachio (sp)

L17 studies (sp)

Answer: all changed

Introduction.

L108 and L126 section 2.2. is repeated twice

Answer: Our fault. Changed in the Results section

Not sure if authors had previously hypothesized on outcomes of research questions – might be worthwhile including hypothesis for this study

Answer: modified the Introduction to try and improve the clarity of the objectives.

Results.

Figure 3 and 4- based on differences in expression pattern of aflR and aflD across climate-related abiotic factors, it is clear that there are genetic differences in AB3 and AB10 – but you don’t see significant chances in AFB1 levels between both strains – explanation?

Answer: Molecular identification was done and compared to a type strain (NRRL one). They are thus both A. flavus strains. However, previous studies on the water and temperature relations showed that generally they behaved similarly in terms of growth on pistachio-based media and on raw pistachio nuts. AFs production may depend on the nutritional media used. On conducive defined media these strains produce high amounts of AFB1. However, on the pistachio nut-based media consisting of a lipid rich nutritional matrix they produce different amounts of AFB1. Indeed, the previous study showed that on raw pistachios much less AFB1 was produced (Baazeem et al., 2021).  

Table 1. Sequencing results of isolated strains and type strain using ITS1 & 2 and ITS3 & 4 primer pairs for molecular identification of strains AB3 and AB10. *Type strain from the Agricultural Research Services Laboratories of the US Department of Agriculture USDA, New Orleans).

Strain ID          GenBank ID                             Genus              Species            similarity (%)

                        ITS1 & 2           ITS 3 & 4                     

_________        _________        _________        ________          ______  ________

NRRL 3357*      M1204.653        BP4                  Aspergillus        flavus               100/99

AB3                  A4S3_13           SCAU-F-142      Aspergillus        flavus               100/98 

AB10                M1204.653        LPSC 1183        Aspergillus        flavus               100/99

Figure 3. The difference in expression of aflR and aflD is interesting (I wonder whether the difference is due to time of RNA extraction, 10 days); typically wouldn’t you expect expression of nor-1 and aflR be in the same direction? See comment on time dependent expression L363; authors should discuss this to help explain gene expression data

Answer: This is indeed a good point. We agree that on defined media, aflD and aflR expression can be switched on relatively early and within 48-72 hrs. As the aflD gene expression is early in biosynthetic process, after 10 days the sequential activity of this gene was probably completed and later genes including the regulatory gene were still active. Our previous studies on defined media and on maize suggests that perhaps 5-7 days would have been ore appropriate. A kinetic study would certainly provide more useful information in this respect.

Discussion. N/A

Materials and Methods.

L296 – I do not see supp. Table 1

Answer: Should have been available. It was added.

L313 – how genetically similar/different AB3 v. AB10 is?

Answer: Ecologically relatively similar. Molecular appear to be similar although in terms of biosynthetic genes – we did not examine relative differences in expression. They certainly – based on the aflD/aflR data respond slightly differently.

L338 -  does gamma radiation at 12-15 kGys impact fungi?

Answer: This level of gamma irradiation certainly kills both phyllosphere and internal fungi present. Previous studies have shown that this level of irradiation does this but retains the germinative capacity of wheat/maize/oats. Thus we use this method because autoclaving is to extreme and changes the nature of the substrate.

L363 – ten day incubation – is this optimal time for aflR and aflD gene expression levels?  In standard rich liquid media (for example, YES) we know that aflR peaks between 48-72 hours, do authors know what is the time dependent gene expression level in PNA? I believe this is important for the interpretation of gene expression data.

Answer: We agree. Early measurements would have been better and a kinetic study would have helped here. Some preliminary studies were carried out but after 5 days only.

L384 – I would suggest that authors keep these genes consistent, perhaps Nortaq-1/nortaq2 (aflD)? OK, I see this in Table 1.

Answer: We have tried to use aflD

Reviewer 3 Report

The article „Impacts of climate change interacting abiotic factors on growth, aflD and aflR gene expression and aflatoxin B1 production by Aspergillus flavus strains in vitro and on pistachio nuts“ is well prepared and designed experiment. There are several necessary corrections and changes needed prior to the publication to increase reach to readers. 

Line 6: change funfi to fungi

Line 7: please use italic when naming microorganisms (Aspergillus flavus)

Line 11: in text authors are using three different signs tor degree (see line 11, line 19, and line 216), please check the whole manuscript and use one symbol for a degree (°)

Line 29: please change keyword from “gene expression” to “aflR, aflD

Line 40: please remove extra space after the end of a sentence

Line 45: please add a reference for class 1a carcinogen (IARC report form 2012),

Line 47-49: please rephrase and shorten the sentence: “Previously, because of this…..”

Line 53: please add reference at end of a sentence

Line 67-70: please provide reference supporting claims in this sentence

Line 72-73: please change the word “optimum” to “optimal”

Line 89: please add the effect of light on aflatoxin production as one of the possible interactions (http://dx.doi.org/10.3390/toxins10120528; https://doi.org/10.1007/s10267-006-0336-2), and also fluctuating day/night temperature that have affected OTA production (reference 17)

Line 103: Please explain abb. on first usage (PNA)

Line 107: please change number 2 to index in CO2

Figures: please add error bars with Standard deviation, standard error of means,  or interquartile range (depending on data distribution)

Please calculate and comment correlations or add correlation graphs/matrices for AFB1 (in PNA and pistachio) vs gene expression (aflR and aflD) and radial growth. They can be added also as supplementary materials.

Figure 2 AFB1 in strain AB10 at 37°C – please check the markings of LSD test on the figure since they all look like they are similar in AFB1 amount, and yet they are statistically different. Maybe it is a typo.

Fig 3 & 4 can you please explain additionally in text how you get different responses of gene expression to the same conditions between AB3 and AB10 strains?

Line 184. Please check the sentence, it seems unfinished (…raw pistachio nuts under.)

Line 214 please check the name of genes in the text due to different naming

Line 229 please close the bracket

Conclusions: please avoid the usage of the word “some” in the conclusion, rather use exact numbers.

Line 282 – 289: please in conclusion leave only conclusions from your study, and other suggestions should be in the discussion. So please exclude this last part of the conclusion (form sentence “In terms of impacts of….” Till the end of the conclusion, and return it to the discussion.

Line 305 – please add a volume of glycerol that was used

Line 309 – please use uniformly abb for volume (litre) you are mixing in manuscript “l” and “L”. Both of them are allowed, but only one should be used in the whole manuscript (line 309,320,321…: ml; line 342,375,380… mL)

Line 333: did you check the CO2 levels before flushing with 1000 ppm CO2, since the concentration of CO2 can be changed due to microbial growth, so the final concentration might be higher?

Line 349: please explain used abb: m.c.)

Line 356 – please explain how did you centrally inoculated the Petri plates: using liquid inoculum (how much), using agar plugs (diameter), or inoculation loop?

Line 358: please indicate how many replicates (2-10)

Line 368: please indicate did you work with wet biomass, dried biomass, or freeze-dried biomass

Line 369: please don’t start a sentence with a number

Line 374: please change RPM to RCF (x g)

Line 396: please add reverse transcription conditions (temperature/enzymes/volumes)

Line 397: please use italic for gene names

Line 425: you don’t need to explain that calculations were performed by Microsoft Excel®

Line 442: what was the weight of agar plugs used for analysis? Also, what was the position of agar plugs that were analysed (random, centre and sides, three spots on edge of culture???). This is important since you have different concentrations of aflatoxins depending on the age of culture (middle area should have highest levels, while edge should have lower levels)

Please add the RPM of extraction on KS 501

Line 465: please explain the drying procedure and exposure to sunlight (that could reduce the AFB1 concentrations).

Line 467: Please explain how was recovery calculated and did you correct the results for recovery?

Line 468: please add blender type and used speed

Line 469: please add PBS pH and ion molarity

Line 473: overnight drying was at room temperature or elevated temperature?

Statistical analysis: please explain what test did you use to confirm Homoscedasticity (as one of the prerequisites for ANOVA).

Author Response

All changes highlighted blue text.

The article „Impacts of climate change interacting abiotic factors on growth, aflD and aflR gene expression and aflatoxin B1 production by Aspergillus flavus strains in vitro and on pistachio nuts“ is well prepared and designed experiment. There are several necessary corrections and changes needed prior to the publication to increase reach to readers.

Line 6: change funfi to fungi

Line 7: please use italic when naming microorganisms (Aspergillus flavus)

Line 11: in text authors are using three different signs tor degree (see line 11, line 19, and line 216), please check the whole manuscript and use one symbol for a degree (°)

Line 29: please change keyword from “gene expression” to “aflR, aflD”

Line 40: please remove extra space after the end of a sentence

Line 45: please add a reference for class 1a carcinogen (IARC report form 2012),

Line 47-49: please rephrase and shorten the sentence: “Previously, because of this…..”

Line 53: please add reference at end of a sentence

Line 67-70: please provide reference supporting claims in this sentence

Line 72-73: please change the word “optimum” to “optimal”

Answer: Thanks. All the above have hopefully been addressed now in the revised version

Line 89: please add the effect of light on aflatoxin production as one of the possible interactions (http://dx.doi.org/10.3390/toxins10120528; https://doi.org/10.1007/s10267-006-0336-2), and also fluctuating day/night temperature that have affected OTA production (reference 17)

Answer: Now added these references. We agree.

Line 103: Please explain abb. on first usage (PNA)

Answer: added

Line 107: please change number 2 to index in CO2

Answer: Done

Figures: please add error bars with Standard deviation, standard error of means,  or interquartile range (depending on data distribution)

Answer: We have now added the bars for LSD values. We could add the SEs if these are absolutely necessary. We think this would make the Figures look rather messy??

Please calculate and comment correlations or add correlation graphs/matrices for AFB1 (in PNA and pistachio) vs gene expression (aflR and aflD) and radial growth. They can be added also as supplementary materials.

Answer:

Figure 2 AFB1 in strain AB10 at 37°C – please check the markings of LSD test on the figure since they all look like they are similar in AFB1 amount, and yet they are statistically different. Maybe it is a typo.

Answer: Checked and modified

Fig 3 & 4 can you please explain additionally in text how you get different responses of gene expression to the same conditions between AB3 and AB10 strains?

Answer: Strains often react similarly to interacting conditions in terms of growth. However, while both strains were molecularly identified and compared to a type strain (NRRL strain) they will of course have some genetic differences. We believe that it would have been better to have data after 4-5 days as opposed to 10 days as aflD is an early biosynthetic gene in the pathway and may have already exhibited maximum expression as has been found in our previous work on defined media and on maize kernels (Medina et al., 2015, 2017).. 

Line 184. Please check the sentence, it seems unfinished (…raw pistachio nuts under.)

Answer: Our mistake – under deleted.

Line 214 please check the name of genes in the text due to different naming

Answer: checked

Line 229 please close the bracket

Answer: Done

Conclusions: please avoid the usage of the word “some” in the conclusion, rather use exact numbers.

Answer: Modified accordingly.

Line 282 – 289: please in conclusion leave only conclusions from your study, and other suggestions should be in the discussion. So please exclude this last part of the conclusion (form sentence “In terms of impacts of….” Till the end of the conclusion, and return it to the discussion.

Answer: Done

Line 305 – please add a volume of glycerol that was used

Answer: Done

Line 309 – please use uniformly abb for volume (litre) you are mixing in manuscript “l” and “L”. Both of them are allowed, but only one should be used in the whole manuscript (line 309,320,321…: ml; line 342,375,380… mL)

Answer: Done.

Line 333: did you check the CO2 levels before flushing with 1000 ppm CO2, since the concentration of CO2 can be changed due to microbial growth, so the final concentration might be higher?

Answer: Some trial measurements were done using GC analysis. A sentence has now been added.

Line 349: please explain used abb: m.c.)

Answer: Done

Line 356 – please explain how did you centrally inoculated the Petri plates: using liquid inoculum (how much), using agar plugs (diameter), or inoculation loop?

Answer: Used agar plugs (4 mm) of germinating spores on PNA medium. Details now added.

Line 358: please indicate how many replicates (2-10)

Answer: 3 replicates and carried out twice. This is mentioned.

Line 368: please indicate did you work with wet biomass, dried biomass, or freeze-dried biomass

Answer:

Line 369: please don’t start a sentence with a number

Answer: changed

Line 374: please change RPM to RCF (x g)

Answer: done

Line 396: please add reverse transcription conditions (temperature/enzymes/volumes)

Answer: Now added

Line 397: please use italic for gene names

Answer: Done

Line 425: you don’t need to explain that calculations were performed by Microsoft Excel®

Answer: deleted

Line 442: what was the weight of agar plugs used for analysis? Also, what was the position of agar plugs that were analysed (random, centre and sides, three spots on edge of culture???). This is important since you have different concentrations of aflatoxins depending on the age of culture (middle area should have highest levels, while edge should have lower levels)

Answer: Plugs were taken across the colonies to ensure we use a representative plugs of the colony. This is clearly stated.

Please add the RPM of extraction on KS 501

Answer: Done

Line 465: please explain the drying procedure and exposure to sunlight (that could reduce the AFB1 concentrations).

Answer: Done. Drying oven at 50oC in the dark.

Line 467: Please explain how was recovery calculated and did you correct the results for recovery?

Answer: Yes. Correction was done for recovery.

Line 468: please add blender type and used speed

Answer: Added

Line 469: please add PBS pH and ion molarity

Answer: Done

Line 473: overnight drying was at room temperature or elevated temperature?

Answer: Done

Statistical analysis: please explain what test did you use to confirm Homoscedasticity (as one of the prerequisites for ANOVA).

Answer: Added.

Round 2

Reviewer 1 Report

This revised version of the manuscript is much improved and reads well; with a few typos that can easily be fixed. My only concern is that the authors didn’t address the suggestion to run a factorial analysis to determine what are the relative effects of each of the factors in study. Otherwise, the manuscript is ready for publication. I am attaching a revised version with a few typos (in red font) that need to be addressed.  

Author Response

We have made all minor corrections recommended.

We have used the LSD values based on the three way interaction of the factors. In addition I think there is already significant information on two way interacting factors. This has included the three-way interacting factors.

Reviewer 3 Report

Authors have corrected the manuscript according to the suggestions, and the paper can be published

Author Response

Thank you for your comments. The changes recommended have certainly improved the manuscript significantly (first round). We have now also made the 2nd round minor corrections. These are again highlighted in blue text.